# Visualizing omicron: COVID-19 deaths vs. cases over time

**Ramy Arnaout[1,2]\*, Rima Arnaout[3]**

**1** Division of Clinical Pathology, Department of Pathology, Beth Israel Deaconess Medical Center, Harvard Medical School, Boston, MA, United States of America, **2** Division of Clinical Informatics, Department of Medicine, Beth Israel Deaconess Medical Center, Harvard Medical School, Boston, MA, United States of America, **3** Division of Cardiology, Department of Medicine, Bakar Computational Health Sciences Institute, Chan Zuckerberg Biohub Intercampus Research Award Investigator, Biological and Medical Informatics Graduate Program, University of California, San Francisco, San Francisco, CA, United States of America

\* rarnaout@bidmc.harvard.edu

**Data Availability Statement:** Data and code are freely available via the rarnaout/Covidcycles repository (https://github.com/rarnaout/Covidcycles). Data is also available at the following: https://github.com/owid/covid-19-data/raw/master/public/data/jhu/biweeklycasespermillion.csv

## Abstract

For most of the COVID-19 pandemic, the daily focus has been on the number of cases, and secondarily, deaths. The most recent wave was caused by the omicron variant, first identified at the end of 2021 and the dominant variant through the first part of 2022. South Africa, one of the first countries to experience and report data regarding omicron (variant 21.K), reported far fewer deaths, even as the number of reported cases rapidly eclipsed previous peaks. However, as the omicron wave has progressed, time series show that it has been markedly different from prior waves. To more readily visualize the dynamics of cases and deaths, it is natural to plot deaths per million against cases per million. Unlike the time-series plots of cases or deaths that have become daily features of pandemic updates during the pandemic, which have time as the x-axis, in a plot of deaths vs. cases, time is implicit, and is indicated in relation to the starting point. Here we present and briefly examine such plots from a number of countries and from the world as a whole, illustrating how they summarize features of the pandemic in ways that illustrate how, in most places, the omicron wave is very different from those that came before. Code for generating these plots for any country is provided in an automatically updating GitHub repository.

## Introduction

Visualization is an essential tool for summarizing and making sense of data. During the COVID-19 pandemic, time-series plots of cases and deaths have become fixtures of news reports, social media posts, and dashboards [1–3]. In time-series plots, the x-axis is time and the y-axis is the variable of interest, for example daily new cases per million population. Information about a second variable of interest can be presented as a second line plotted against the same x-axis, with or without a secondary y-axis, but this is not the only way to present two variables. For example, if the desire is to draw the viewer's attention to the ratio of the two variables, the ratio can be plotted over time; however, the absolute magnitudes are lost if only the ratio is plotted.

https://github.com/owid/covid-19-data/raw/master/
public/data/jhu/biweeklydeathspermillion.csv
https://ourworldindata.org/covid-vaccinations
https://covariants.org/per-country https://www.
gisaid.org/.

**Funding:** The authors (R.A. and R.A.) were funded
by the NIH (1R01AI148747 and 1R01HL150394)
and the Gordon and Betty Moore Foundation. The
funders had no role in study design, data collection
and analysis, decision to publish, or preparation of
the manuscript.

**Competing interests:** The authors have declared
that no competing interests exist.

An alternative visualization approach is to plot the two variables against each other as a scatterplot, one on the x-axis, the other on the y-. In such a plot, time has no axis of its own, but can be represented visually as a gradient of weight, width, or color along the line, and/or can be conveyed using arrows or arrowheads along the line (similar to how vector fields are often displayed). Such plots are standard in the study of dynamical systems, leading for example to the phase diagrams of differential equations [4, 5]. They are useful for drawing attention to commonalities and differences in the relationship between a pair of variables over time. In this spirit, we plotted day-by-day deaths vs. cases for the COVID-19 pandemic to date, to see what such plots might illustrate about the pandemic, and possibly its near-term tendencies.

## Materials and methods

### Search

Image searches for "COVID-19 cases vs. deaths scatterplot" and simple variants thereof were performed on the Google, DuckDuckGo, and Brave search engines on January 10, 2022. For each search engine, the number of the first 20 results that (1) were plots of both cases and deaths (2) as a function of each other (as opposed to over time) (3) at multiple time points was recorded.

### Data

Biweekly cases per million and biweekly deaths per million were taken from the Our World in Data GitHub repository [6] via the following URLs: https://github.com/owid/covid-19-data/raw/master/public/data/jhu/biweeklycasespermillion.csv and https://github.com/owid/covid-19-data/raw/master/public/data/jhu/biweeklydeathspermillion.csv. These biweekly moving averages are updated daily. (Our World In Data in turn credits the Center for Systems Science and Engineering (CSSE) at Johns Hopkins University, which itself lists numerous additional sources.) Vaccination rates as of January 9, 2022 were also via Our World in Data (https://ourworldindata.org/covid-vaccinations). Variants over time were via CoVariants (https://covariants.org/per-country) [7], which in turn comes from the GISAID Initiative (https://www.gisaid.org/). All data were accessed on January 24, 2022.

### Code

Plots were made using Python (see Availability below) and OmniGraffle 7.19.2 on an Apple M1 MacBook Air (2021) and updates on GitHub.

### Availability

Data and code are freely available via the rarnaout/Covidcycles repository (https://github.com/rarnaout/Covidcycles).

## Results

### Few plots of deaths vs. cases over time during the COVID-19 pandemic

Although plotting two variables against each other over time is a well established practice (as mentioned in the Introduction), plotting deaths vs. cases over time appears to be uncommon during the COVID-19 pandemic (**Table 1**). Web searches revealed occasional plots of total deaths vs. total cases, but only two plot of deaths vs. cases over time: one for a one-month window from April 2020 posted by the user prograft to the COVID-19 Data Visualizations subreddit [8], and a second from an RStudio blog (titled "Not Useful") [9, 10].

**Table 1. Images of COVID-19 deaths vs. cases over time, various search engines.**

| Source | Plot both cases and deaths... | ...as a function of each other... | ...at multiple timepoints |
|---|---|---|---|
| Google | 7 | 7 | 1 |
| DuckDuckGo | 16 | 2 | 1 |
| Brave | 15 | 1 | 0 |

### Plots of death vs. cases illustrate country-specific COVID-19 waves

We plotted deaths per million vs. cases per million from the start of the pandemic until the time of writing, for 16 countries, colored by wave (please note that both x- and y- axes may be scaled differently from graph to graph). We comment on several examples.

**South Africa (Fig 1A).** The plot for South Africa clearly illustrates the main pattern of counterclockwise loops (Fig 1A). Each loop describes a wave of the pandemic. In each loop, cases rise, then deaths rise, then cases fall, and finally deaths fall. The peak number of cases is always to the right (the date label indicates the peak for each wave). The wave ends with the population back near the origin of the plot, with few cases and few deaths. The waves occurred roughly every six months, with two mid-summer and two mid-winter peaks.

South Africa's first wave (purple) reached its peak number of cases on July 14, 2020 (labeled). The start of the second wave (blue) coincided with the appearance of the SARS-- CoV-2 beta variant, and indeed beta dominated this wave, with its frequency in the population reaching its peak frequency (0.97) nearly coincident with the peak in cases. The third wave (green) was similarly dominated by the delta variant, and the fourth (red) dominated by omicron. For the beta, delta, and now omicron waves, reported cases peaked around 4,000–5,000 cases per million.

It is interesting to note that the blue (beta), green (delta), and red (omicron) loops get progressively flatter: this indicates falling case mortality, from 125 cases per million for beta to only around 25 cases per million for omicron. Thus in South Africa, the beta wave was deadlier per capita than the delta wave, even though the delta variant is known to be quite virulent and led to substantial mortality around the world. Also, note that each wave reaches further to the right. Thus, over the last three waves, the virus has been de facto less virulent but more infectious.

It is tempting to attribute this pattern at least in part to herd immunity from natural infection and not vaccination, since only roughly a quarter of the South African population was fully vaccinated as of early January 2022. Otherwise, it is reasonable to expect to have seen

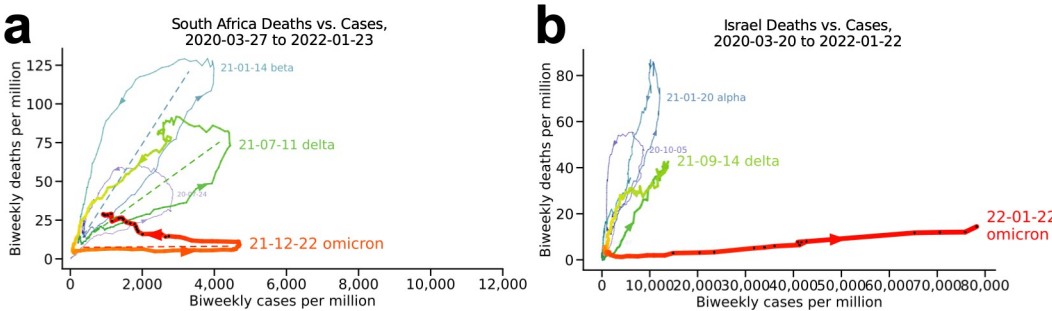

**Fig 1. South Africa and Israel.** All plots are colored by wave. Waves are labeled by variant if a single known variant predominated. Date labels appear at the peak cases-per-million for each wave. Arrowheads indicate the flow of time. Dotted lines in (a) are guides to the eye regarding the orientation of the loops, indicating the progressive flattening of the last three waves.

higher mortality for the delta wave. Note that herd immunity is unlikely to be the entire explanation, since laboratory investigation has shown that omicron is inherently both less pathogenic and more transmissible than delta, regardless of a person's prior vaccination or infection status [10–13]. In general for any country, the explanation for the dynamics observed in these plots is almost certainly similarly multifactorial, with reporting, vaccination status, prior covid infection, policy (e.g. lockdowns), demographics, and health status all contributing.

**Israel (Fig 1B).** Israel, a more vaccinated country (64% fully vaccinated as of January 10, 2022, meaning they have received a single-dose vaccine or both doses of a two-dose vaccine), provides an interesting comparison to South Africa (Fig 1B). It also had four main waves, with the second (blue) outpacing the first (purple) in both cases and deaths. (Israel's second wave was caused by the alpha variant.) As in South Africa, Israel's delta wave (green) also had more daily cases per million but fewer deaths per million, and its fourth main wave (red), the omicron wave, set a new record for daily cases per million but without a notable rise in deaths per million so far. The case peaks for Israel's previous main waves all came a week to a few months after South Africa's. It remains to be seen whether Israel's fourth wave will follow the South African pattern, as it has so far.

**Italy, Denmark, Sweden, and the United Kingdom (Fig 2).** These European countries show variations on a different pattern from that seen in South Africa and Israel: a highly deadly wave in the end of 2020 and the start of 2021, with a prominent double peak visible as a loop within a loop in Italy, the United Kingdom, and Sweden (Fig 2A, 2C and 2D). In all cases, this coincided with the alpha variant displacing the previously predominant 20E/EU1 strain and rising to near fixation. The similarity of the pattern is notable given the policy differences among these countries in how to address the pandemic, especially in Sweden. For Sweden, the unusual spikiness of the January 2021 loop is likely a reporting artifact.

In Denmark (Fig 2B), the abrupt inflection point as the curve turns from orange to red coincided with the rapid shift from the delta variant, a highly virulent strain, to omicron, which biomedical research suggests is less pathogenic [14, 15] despite antibody escape [16–18], especially in previously exposed/highly vaccinated populations [18, 19] (Denmark is 80% fully vaccinated), consistent with the South African experience. The plot for Denmark is thus

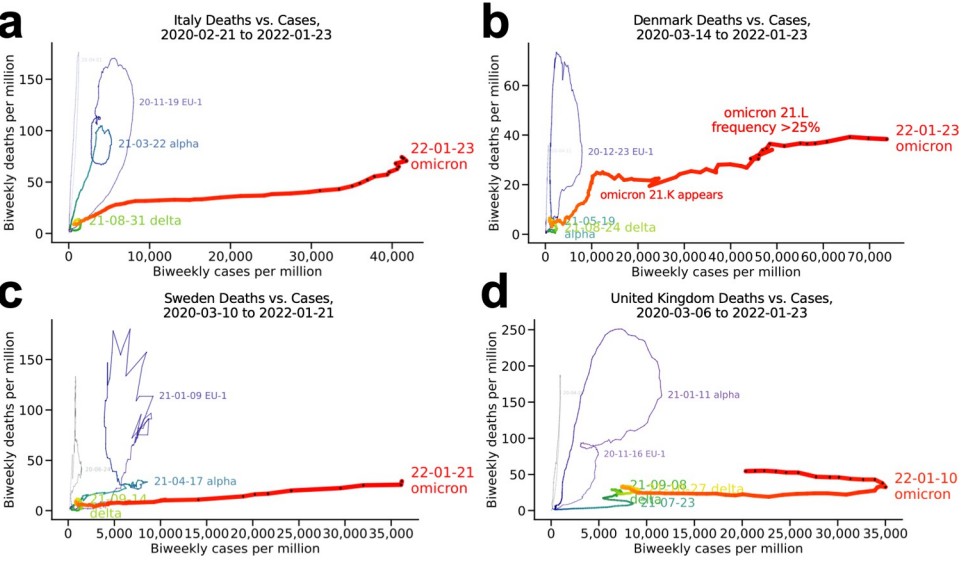

**Fig 2. Select European countries.** (a) Italy, (b) Denmark, (c) Sweden, and (d) United Kingdom.

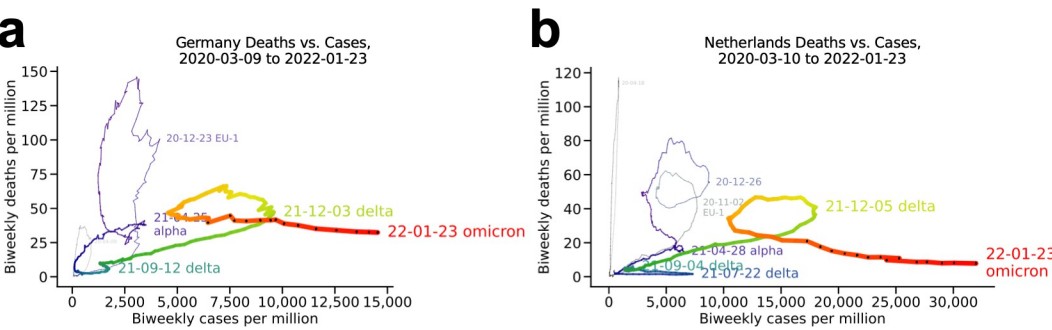

**Fig 3. More European countries: Germany and the Netherlands.** (a) Germany and (b) the Netherlands.

interpreted as a "loop-pulled-taut" around the inflection point, with a somewhat similar (though less pronounced) effect in Italy. This is consistent with the temporary fall of deaths-per-million as cases rise, before the rise again as the omicron loop continues. The slight uptick coincided with expansion of the omicron 21.L variant, which had reached a prevalence of approximately 50% by January 24, 2022 [7].

Note that even with the compression of the x scale in these plots, relative to the plots for South Africa and Israel above, the dynamics—i.e. the loop patterns—of these European countries have been different. The reason for the compression is of course the great number of omicron cases (red), which for each of these countries dwarfs the daily cases per million seen previously. Note also that the y scales are fairly comparable. The density of the black dots indicates the omicron wave peaking. However, in every case it has begun according to a very different trajectory from the previous waves.

**Germany and the Netherlands (Fig 3).** These neighboring countries share the double loop of their fellow European countries (Fig 2) but are notable for an additional loop in the final months of 2021 and the start of 2022 (red). In both cases, the delta variant still accounted for between half and 90% of cases through the period shown. The wave seen subsiding in these plots merged into the omicron wave, creating another loop within a loop as the omicron wave hits these countries. Compare to the plot for Denmark (**Fig 2B**).

**Japan and South Korea (Fig 4).** Japan (**Fig 4A**) barely completed a wave dominated by the alpha variant (blue) before entering its delta-variant wave (green). Although January 2022 reporting [7] for nearby South Korea (**Fig 4B**) indicates it is still in its delta wave, transitioning from delta 21I to delta 21J, the trajectory of its latest wave (red) is consistent with omicron having begin to have a major effect. Note the small absolute numbers in both these countries.

**India and China (Fig 5).** The plots for the two most populous countries look very different from each other. By the time of this writing, India's relatively prominent delta wave

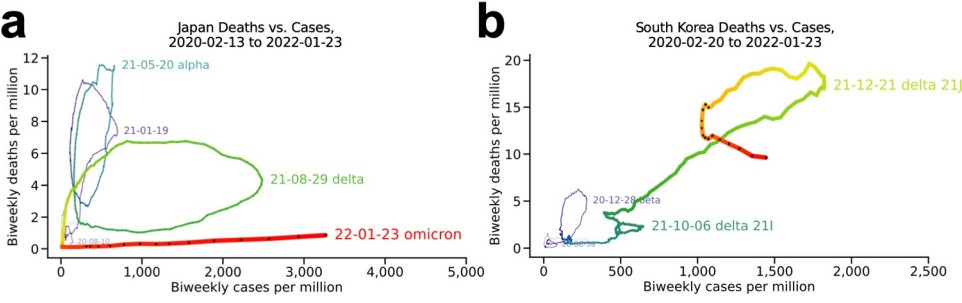

**Fig 4. Japan and South Korea.** (a) Japan and (b) South Korea.

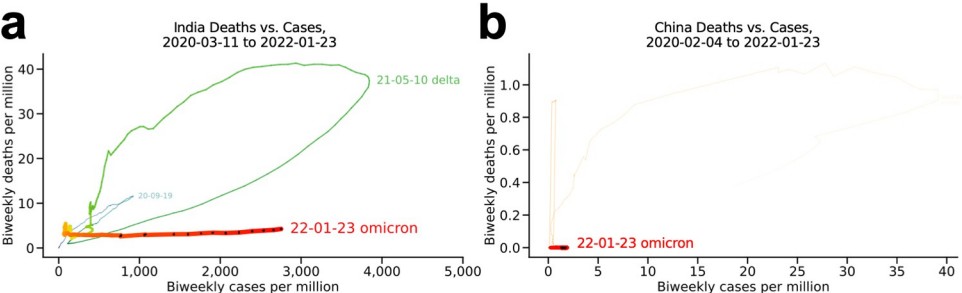

**Fig 5. India and China.** (a) India and (b) China. The two most populous countries.

resolved (green) and its omicron wave had begun (red), not unlike many other countries (**Fig 5A**). China, where the SARS-CoV-2 virus was first reported, has reported almost no cases or deaths per million since early in the pandemic (note the axis scales), a notable outlier among the countries presented here (**Fig 5B**).

**Australia and New Zealand (Fig 6).** Australia (**Fig 6A**) is notable for low case and death rates but high mortality per reported case, visible in the steepness of the yellow loop, which was dominated by the delta variant (even on an uncompressed x-scale). In December 2021-January 2022, Australia decided omicron (red) would be uncontainable and changed from a strategy featuring lockdowns to one geared toward blunting the effect on mortality, but its omicron loop, which peaked January 18, has been by far its worst to date. New Zealand (**Fig 6B**), the country with the smallest population in the countries discussed here (5.1 million people, vs. 5.8 million for second-lowest Denmark), has reported very few cases and deaths per million throughout the pandemic, including a very small delta wave (red). The choppiness of the line may be due to these small absolute numbers.

**North America (Fig 7).** The plot for the United States (**Fig 7A**) illustrates a substantial blunting of the alpha wave (blue) relative to the previous wave (purple), but its delta wave (green) was deadlier per case, without the flattening seen for South Africa (**Fig 1A**). The loop for the delta wave was blunter but otherwise similarly shaped to the January, 2021 wave nine months earlier, indicating a similar per-case fatality rate. Like Denmark, Germany, and the Netherlands, the death rate remained high into the start of the omicron wave, but the additional death due to the omicron wave has been minimal so far, and the omicron wave has peaked. Canada (**Fig 7B**) is similar except that its delta wave (green) is shallower than its alpha wave (blue), with the omicron wave beginning shallower still, reminiscent of South Africa. The choppiness around the omicron peak may reflect abandonment of case reporting around the peak. The blunting of successive waves in Mexico (**Fig 7C**) bear an even stronger resemblance

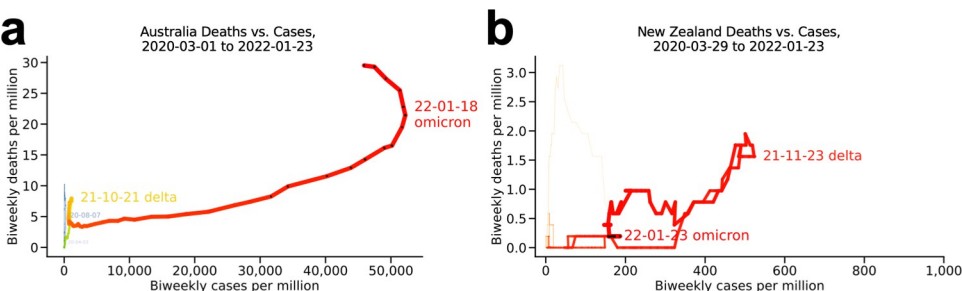

**Fig 6. Australia and New Zealand.** (a) Australia and (b) New Zealand.

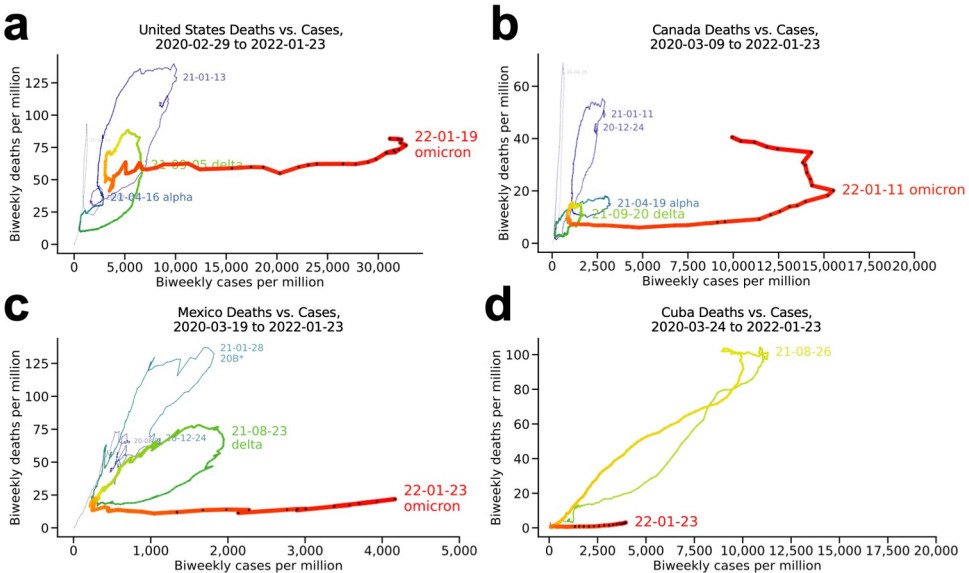

**Fig 7. North America.** (a) United States, (b) Canada, (c) Mexico, where variant *20B accounted for 47% of reported strains in the January 2021 case peak, and (d) Cuba.

to that seen for South Africa and Israel. Mexico reported being 56% fully vaccinated as of January 2022, with Cuba (**Fig 7D**) at 86%, Canada at 78%, and the United States at 62%.

**The world (Fig 8).** Worldwide the picture is of a devastating first wave (purple), a slightly shallower second wave (blue), and blunted third wave (green), and very high cases at the start of the fourth wave, dominated by omicron in almost all countries visualized. New waves have hit every three to five months. Again, the distinctiveness of the omicron wave is clear: very

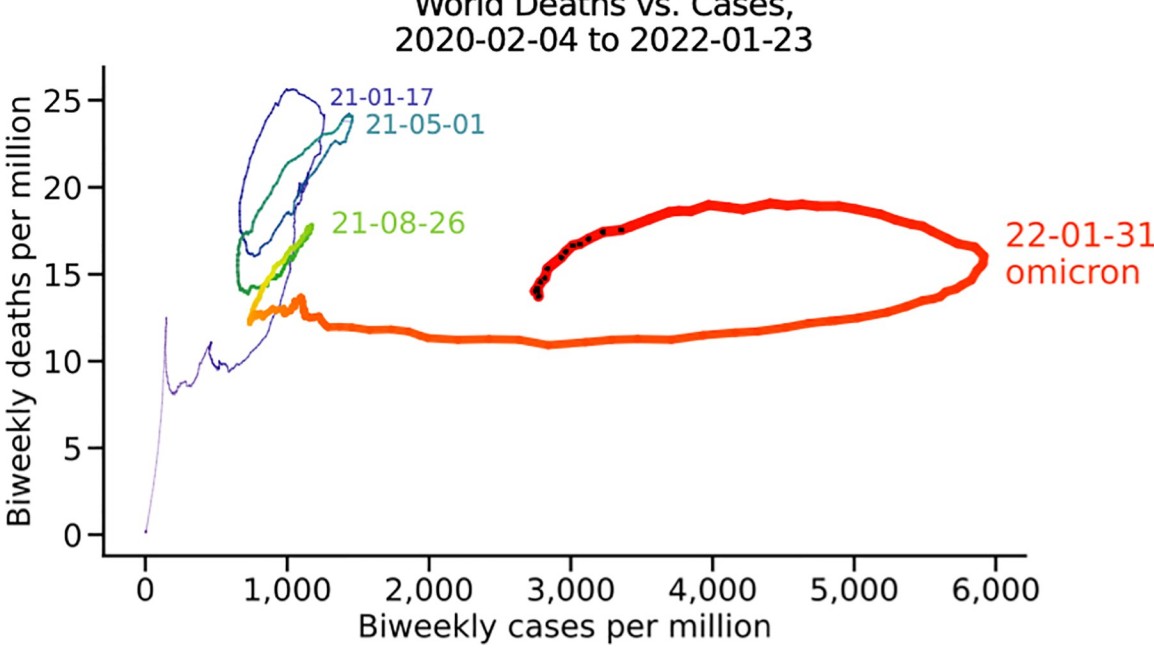

**Fig 8. The world.** All reported cases and deaths.

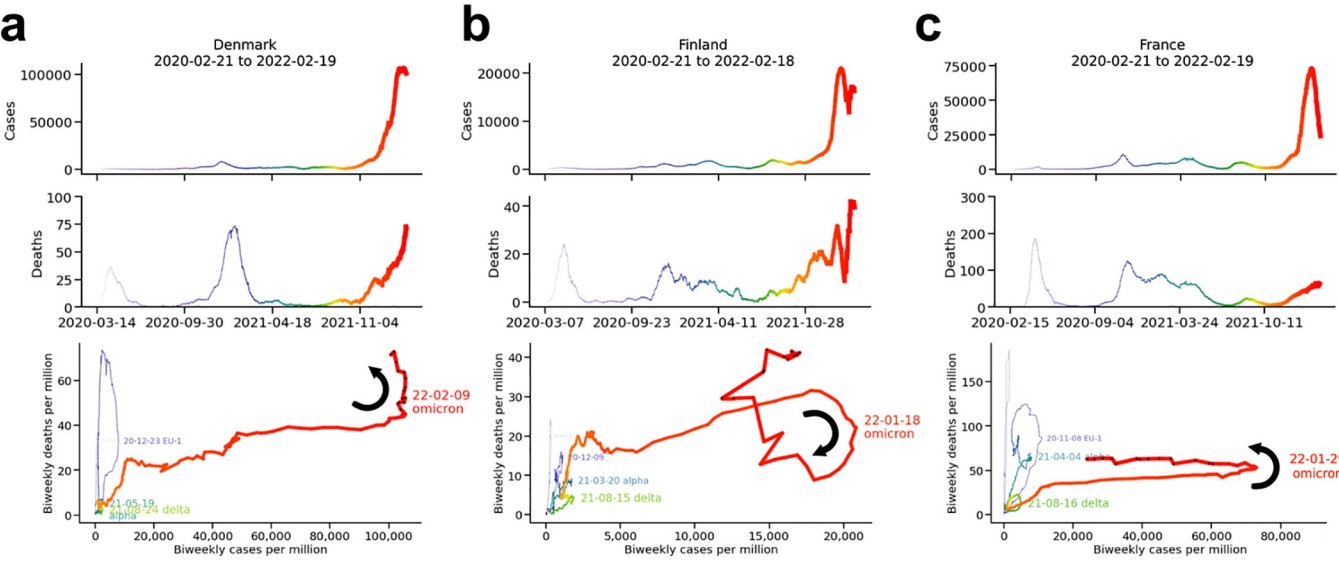

**Fig 9. Time series vs. bivariate plots.** (a) Denmark, (b) Finland, and (c) France.

high cases per million without a substantial rise in death rate per million, at least as of this writing. The density of black dots in the middle of January strongly suggests that the omicron wave will peak worldwide by February 1, 2022.

**Comparison to bivariate plots (Fig 9).** When visualized as univariate time series, the infection dynamics of Denmark, Finland, and France appear similar (top two rows, plotted through mid-February 2022). In contrast, the bivariate plots illustrate substantive differences among the three countries (bottom row), most notably the curious clockwise cycle of Finland's omicron wave (curved arrow), unique in the countries we examined. Contrast to the counter-clockwise cycles of Denmark, a fellow Nordic country, and France (curved arrows).

## Discussion

The x-y scatterplot with time as an implicit variable is not a new invention. However, it has been little used in relation to cases and deaths in the COVID-19 pandemic. This communication may be considered a humble re-introduction, with brief mention of some of the observations that such visualization can facilitate. The most striking observation is the difference of the nascent omicron wave to those that have come before, as well as the ease in visualizing different infection dynamics compared to univariate time series plots. Also evident is a general flattening of subsequent waves, following the initial outbreak in 2020, most evident in South Africa and Israel and to a lesser extent in Canada and Japan. Finally, regional differences and other patterns are clearly visible.

Given the history of this type of plot (for example, its utility in modeling pressure-volume or flow-volume dynamics for cardiopulmonary systems), it is interesting that it has been little used so far in the pandemic. We speculate that the main reason is that because the waves have taken place on the relatively slow timescale of months, and there have been relatively few of them to against which to see patterns ("small *n*"), their potential utility did not emerge until long after time series had become fixtures of COVID-19 reporting. However, going forward there are a number of variations that may prove interesting. These including sub-analysis by age, race, gender, and vaccination status. Substituting excess deaths per million in place of deaths per million is another example. It may also be fruitful to explore how the rich dynamics

illustrated by these plots correlate with, or can be predicted by, reporting, vaccination status, public policy (e.g. lockdowns, worker compensation), and health status.

We used biweekly cases and deaths, i.e. running averages, to better visualize trends without the noisiness and sampling granularity of daily reporting (e.g. weekend dips). The reader's attention has been called to the differences in scale between the plots, which is sometimes worthy of note. The plots could also have been presented on the same scale, although differences in rates of testing and reporting, known to differ across countries, would have to be taken into account for fair conclusions to be drawn.

In conclusion, plotting deaths per million vs. cases per million over time, with appropriate annotation, is a potentially useful way to visualize waves of infection in the COVID-19 pandemic.

## Acknowledgments

This work would not have been possible without expert data collection and curation at all levels, together with the selfless commitment to openness and sharing that resulted in this data being publicly available.

## Author Contributions

**Conceptualization:** Ramy Arnaout.

**Formal analysis:** Ramy Arnaout, Rima Arnaout.

**Project administration:** Rima Arnaout.

**Software:** Ramy Arnaout.

**Visualization:** Ramy Arnaout.

**Writing – original draft:** Ramy Arnaout, Rima Arnaout.

**Writing – review & editing:** Ramy Arnaout, Rima Arnaout.

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
