## [Decision Letter · Decision Letter 0]

15 Feb 2022

PONE-D-22-02371Visualizing Omicron: COVID-19 Deaths vs. Cases Over TimePLOS ONE

Dear Dr. Arnaout,

Thank you for submitting your manuscript to PLOS ONE. After careful consideration, we feel that it has merit but does not fully meet PLOS ONE’s publication criteria as it currently stands. Therefore, we invite you to submit a revised version of the manuscript that addresses the points raised during the review process. The comments from the reviewers seem minor.

We look forward to receiving your revised manuscript.

Kind regards,

Etsuro Ito

Academic Editor

PLOS ONE

Journal Requirements:

Reviewers' comments:

Reviewer's Responses to Questions

**Comments to the Author**

1. Is the manuscript technically sound, and do the data support the conclusions?

Reviewer #1: Yes

Reviewer #2: Yes

2. Has the statistical analysis been performed appropriately and rigorously? 

Reviewer #1: I Don't Know

Reviewer #2: N/A

3. Have the authors made all data underlying the findings in their manuscript fully available?

Reviewer #1: Yes

Reviewer #2: Yes

4. Is the manuscript presented in an intelligible fashion and written in standard English?

Reviewer #1: Yes

Reviewer #2: Yes

5. Review Comments to the Author

Reviewer #1: This publications accesses publicly available information on COVID cases and deaths and utilizes a novel type of graphic display to illustrate the dynamics of the infection. It includes a large number of fascinating graphs which show differences in the different waves of COVID.

In using the plots to compare countries, as stated in the discussion, it's important to realize that both X and Y axes are scaled differently from graph to graph; I think it would be valuable to say this prior to the detailed country-by-country discussion for the readers' benefit.

I wonder if it's possible to make the discussion more specific about how to utilize this type of graph to illuminate causative epidemiological variables? As a non-specialist in statistics, I'm left a bit unsatisfied by the assertion in the Discussion that subanalyses and different variables may prove fruitful. Why? What sorts of relationships would these plots illuminate that other approaches to data analysis might miss? Can you draw specific analogies to their use in other dynamical systems?

Would it make sense to pick a country or two, display cases vs deaths graphs side-by-side with more-conventional representations, and discuss the differences in information conveyed? Right now, the paper seems slightly epiphenomenal; these are cool graphs, but it's a bit unclear exactly what makes them distinctively useful.

Reviewer #2: In this paper, the authors have utilized an uncommon COVID-19 graph plotting system for visualizing the pandemic data, which however is a standard plotting method used in dynamic systems, to easily visualize and effectively draw attention to commonalities and differences in the relationship between COVID-19 cases and deaths over time for varying waves of COVID-19 variants, particularly the omicron, in a number of countries and the world as a whole. The graph is plotted with the COVID-19 biweekly deaths per million on the Y-axis and biweekly cases per million on the X-axis over time in a scatterplot to visually distinguishes waves seen with each of the variants and to illustrate the rich infection dynamics of the pandemic in different countries.

I see no major issues and one minor issue. On page 4, results section, 1st paragraph, and 2nd sentence, It looks as though the author wanted to state "total death vs. total cases" and not "total death vs. total death". Overall, I accept the publication with one minor corrections since the paper meets all seven criteria set by the journal.

6. PLOS authors have the option to publish the peer review history of their article (what does this mean?). If published, this will include your full peer review and any attached files.

Reviewer #1: **Yes: **Sheldon Campbell

Reviewer #2: No

---

## [Author Response · Author response to Decision Letter 0]

23 Feb 2022

We thank the Editor and the Reviewers for their constructive comments. Please find point-by-point responses below.

Journal Requirements:

 We have made the style updates.

2. PLOS requires an ORCID ID for the corresponding author in Editorial Manager on papers submitted after December 6th, 2016. Please ensure that you have an ORCID iD and that it is validated in Editorial Manager.

Both authors have registered our ORCID iDs in PLOS.

3. Please review your reference list to ensure that it is complete and correct…Any changes to the reference list should be mentioned in the rebuttal letter that accompanies your revised manuscript.

We have reviewed the reference list. It appears there is a newer citation replacing:

Chan MCW, Hui KP, Ho J, Cheung M, Ng K, Ching R, et al. SARS-CoV-2 Omicron variant replication in human respiratory tract ex vivo 2022. https://doi.org/10.21203/rs.3.rs-1189219/v1.

with:

Hui KPY, Ho JCW, Cheung M-C, Ng K-C, Ching RHH, Lai K-L, et al. SARS-CoV-2 Omicron variant replication in human bronchus and lung ex vivo. Nature. 2022. doi:10.1038/s41586-022-04479-6

We have removed (not required): 

Cameroni E, Saliba C, Bowen JE, Rosen LE, Culap K, Pinto D, et al. Broadly neutralizing antibodies overcome SARS-CoV-2 Omicron antigenic shift. 2021.

Reviewers' comments:

Reviewer #1:

“This publications accesses publicly available information on COVID cases and deaths and utilizes a novel type of graphic display to illustrate the dynamics of the infection. It includes a large number of fascinating graphs which show differences in the different waves of COVID.”

Thank you. We agree that this is an interesting and useful way to visualize the dynamics of severity and transmissibility of infection.

“In using the plots to compare countries, as stated in the discussion, it's important to realize that both X and Y axes are scaled differently from graph to graph; I think it would be valuable to say this prior to the detailed country-by-country discussion for the readers' benefit.”

 We have now noted this prior to the country-by-country results.

“I wonder if it's possible to make the discussion more specific about how to utilize this type of graph to illuminate causative epidemiological variables…Would it make sense to pick a country or two, display cases vs deaths graphs side-by-side with more-conventional representations, and discuss the differences in information conveyed? Right now, the paper seems slightly epiphenomenal; these are cool graphs, but it's a bit unclear exactly what makes them distinctively useful.”

We appreciate this point and have now included Fig 9, which picks three countries (Denmark, Finland, France) and displays the time series side by side with the bivariate plots, to illustrate the dynamics more clearly/easily visible in bivariate plotting.

Reviewer #2:

In this paper, the authors have utilized an uncommon COVID-19 graph plotting system for visualizing the pandemic data…I see no major issues and one minor issue. On page 4, results section, 1st paragraph, and 2nd sentence, It looks as though the author wanted to state “total death vs. total cases” and not “total death vs. total death.” Overall, I accept the publication with one minor corrections since the paper meets all seven criteria set by the journal.

 Thank you for this summary and for pointing out the typo. We have now fixed it.

---

## [Editor Report · Decision Letter 1]

28 Feb 2022

Visualizing Omicron: COVID-19 Deaths vs. Cases Over Time

PONE-D-22-02371R1

Dear Dr. Arnaout,

We’re pleased to inform you that your manuscript has been judged scientifically suitable for publication and will be formally accepted for publication once it meets all outstanding technical requirements.

Kind regards,

Etsuro Ito

Academic Editor

PLOS ONE

---

## [Editor Report · Acceptance letter]

2 Mar 2022

PONE-D-22-02371R1 

Visualizing Omicron:
COVID-19 Deaths vs. Cases Over Time 

Dear Dr. Arnaout:

I'm pleased to inform you that your manuscript has been deemed suitable for publication in PLOS ONE. Congratulations! Your manuscript is now with our production department. 

Kind regards, 

on behalf of

Prof. Etsuro Ito 

Academic Editor

PLOS ONE